# HG-DCM: History Guided Deep Compartmental Model for Early Stage Pandemic Forecasting

## Abstract

Early-stage pandemic forecasting is fundamentally constrained by a lack of data. When a new pathogen emerges, there is insufficient historical context to calibrate standard epidemiological models. We introduce the History-Guided Deep Compartmental Model (HG-DCM), a framework designed to overcome this scarcity by systematically transferring knowledge from historical pandemics to the current outbreak. Rather than relying solely on the sparse data of an unfolding crisis, HG-DCM leverages a deep learning backbone to extract universal temporal patterns and parameter dynamics from a comprehensive dataset of past global outbreaks. By integrating these historical insights with epidemiological and demographic metadata, our approach infers robust, interpretable parameters for compartmental forecasting even when current data is minimal. Experimental results on early-stage COVID-19 tasks demonstrate that leveraging historical guidance significantly reduces overfitting and improves stability compared to standard compartmental models and data-isolated deep learning approaches. HG-DCM establishes a new paradigm for pandemic modeling that moves beyond the limitations of single-outbreak data by integrating the collective history of global epidemiology.

## 1 Introduction

Pandemics represent one of the most devastating threats to global health and economic stability, causing catastrophic losses throughout human history—from the Bubonic Plague in the 14th century McEvedy and the smallpox outbreaks of the 18th century Eyler, to the recent COVID-19 pandemic Holshue et al.. A consensus exists that early intervention is the most effective strategy for mitigating these crises. Studies have estimated that timely governmental actions, such as restrictions on mass gatherings and mandatory mask-wearing, could have reduced the total mortality from COVID-19 by as much as 90% Piovani et al.; Li et al..

However, these crucial public health measures are inherently costly, imposing significant social and economic burdens. Consequently, interventions must be reserved for situations presenting a serious risk of a major pandemic. This requirement creates a fundamental conundrum: the optimal window for decision-making occurs in the initial, exponential phase of an outbreak. It is precisely during this critical period that data is extremely scarce, with very little reliable information to accurately forecast the future trajectory and severity of the pandemic Shea et al.; Lipsitch et al..

Standard forecasting approaches, particularly compartmental models (e.g., SIR, SEIR variants), struggle severely in this data-poor environment Roda et al.. Because these models fit incidence curves separately for each location using only data that is available for the *current* pandemic, they are highly prone to overfitting initial noise rather than capturing stable underlying transmission trends.

This limitation highlights a key difference between mathematical models and human epidemiological intuition. An experienced epidemiologist can filter out early noise by drawing upon a mental library of historical curves and outbreak dynamics—they possess the memory of how past outbreaks evolved. In contrast, standard computational models treat each new pandemic as an isolated, de novo event. While every pandemic is biologically unique, the macroscopic dynamics of spread—driven by human social behavior, response to interventions, and mobility patterns—often follow universal, discernible patterns observed in past outbreaks Viboud et al.. By failing to leverage this wealth of historical knowledge, current models miss a critical opportunity to stabilize early-stage predictions.

The core innovation of HG-DCM is a method to operationalize historical data from biologically distinct pandemics through a knowledge-transfer mechanism. We utilize a neural network backbone that learns to map early-stage signals and associated metadata (e.g., demographics, healthcare capacity) to the underlying parameters of a standard compartmental model (e.g., DELPHI Li et al.). This allows the model to "fill in the gaps" of missing current data with learned priors regarding parameter evolution derived from history.

A natural question arises regarding why training on biologically distinct diseases assists in forecasting a novel pathogen. We argue that while the biological specifics differ, the macroscopic dynamics of spread are universally constrained by human social behavior and public health responses. HG-DCM does not treat historical data as a ground truth for specific parameter values, but rather as a guide for the dynamics of how these parameters evolve. For example, it learns how transmission rates typically decay over time in response to interventions, a pattern that holds true regardless of the specific virus.

Our approach preserves the epidemiological interpretability of compartmental models while solving their primary weakness: the reliance on sparse initial data.

To our knowledge, this is the first study to develop a forecasting framework that systematically leverages data from multiple prior pandemics to predict the trajectory of a newly emerging one. While previous work has borrowed parameter priors from earlier outbreaks Tindale et al. or transferred models between related epidemics Roster et al., no prior research has integrated information across a wide range of different pandemics.

As part of this effort, we constructed a new, comprehensive pandemic dataset including time-series case and death data, along with associated pandemic- and country-level meta-data, from major global outbreaks since 1990 (e.g., COVID-19 U.S. Department of Health & Human Services (2023); The New York Times (2021); World Health Organization (2023), Ebola Centers for Disease Control and Prevention (2016), SARS World Health Organization (2003); imdevskp (2020), Dengue Nic (2020), and seasonal influenza Centers for Disease Control and Prevention (2023); Our World in Data (2023)).

We applied HG-DCM to the challenging task of early COVID-19 forecasting across 258 global locations, finding that our history-guided approach consistently and significantly outperforms state-of-the-art methods—including the original DELPHI model and advanced deep learning-only models—that rely solely on current data. This study provides strong evidence that integrating historical data into compartmental models through neural network guidance can significantly enhance the accuracy and stability of early pandemic forecasting, yielding a robust tool for public health decision-makers.

## 1.1 RELATED WORK

Our work sits at the intersection of epidemiological modeling and deep learning. While the literature on COVID-19 forecasting is vast, we focus here on methods relevant to the "cold-start" problem: forecasting when data for the target disease is scarce.

**Compartmental and Mechanistic Models**   Standard epidemiological forecasting relies on compartmental models, such as SIR and SEIR, which describe disease spread using differential equations (Ross; Schlickeiser & Kröger; KERR). Advanced variants like the DELPHI model (Li et al.) incorporate realistic factors such as under-detection and government interventions. While highly interpretable, these models depend heavily on accurate parameter initialization. In the early weeks of a new pandemic, determining these parameters is often impossible due to the lack of calibration data, leading to significant overfitting and instability.

**Hybrid Deep Learning Frameworks**   To overcome the rigidity of pure mechanistic models, recent works have proposed hybrid frameworks that fuse neural networks with epidemiological priors. DeepCOVID (Rodriguez et al.) and EiNNs (Rodríguez et al.) demonstrate how deep learning can operationalize real-time signals for robust forecasting. Similarly, DeepGLEAM (Wu et al.) and Neural ODEs (Kosma et al.) successfully integrate mechanical constraints into learning processes to ensure physically consistent predictions. In terms of uncertainty quantification, methods like EpiFNP (Kamarthi et al.) and DSA-BEATS (Motavali et al.) have made significant strides in estimating confidence intervals and handling complex temporal dependencies. While some of these architectures have combined compartmental models with deep learning, they only consider data

from the current pandemic in prediction, and require a sufficient stream of data to learn effective representations. Consequently, they are less applicable to the "cold-start" phase—specifically the first 2 to 8 weeks—where the training signal from the current outbreak is too sparse to train complex backbones like ResNets or Transformers without overfitting.

**Transfer Learning and Agent-Based Approaches**   To address data scarcity, transfer learning has been explored in various forms, though often with different goals than ours. Some approaches focus on spatial transfer, moving knowledge from regions with advanced outbreaks, such as Italy, to those in early stages like the US (Panagopoulos et al.). Other strategies include Agent-Based Models (ABMs), such as Differentiable ABMs (Chopra et al., 2023), which offer granular simulation capabilities but operate on a fundamentally different paradigm requiring detailed mobility and interaction data that is unavailable globally in many less developed countries in early stages.

**Our Contribution: Cross-Disease Temporal Transfer**   Our work addresses the gap left by the methods above. Unlike current models that primarily rely on data from the current pandemic, the HG-DCM framework introduces cross-disease temporal transfer. We operate on the premise that while pathogens differ biologically, the human-driven dynamics of spread share universal patterns across history. By treating historical pandemics, such as Dengue or seasonal Influenza, as a source domain, we can initialize robust forecasting models for a novel pathogen before sufficient single-disease data exists. This allows us to regularize deep compartmental models effectively during the critical "cold-start" window, providing a distinct advantage over single-disease architectures.

## 2 METHODS

We introduce the History-Guided Deep Compartmental Model (HG-DCM), a novel framework designed to enhance early pandemic forecasting by combining the interpretability of traditional epidemiological models with the expressive power of deep learning over historical data. The overall architecture is illustrated in Figure 1.

### 2.1 MODEL CONSTRUCTION

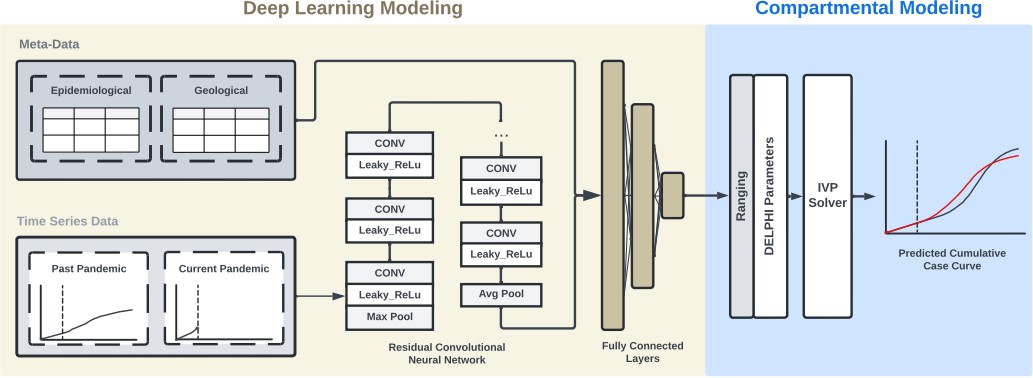

Figure 1: **Model Architecture of HG-DCM** HG-DCM consists of two parts: a deep learning parameter predictor $f(\cdot)$ and a compartmental model $h(\cdot)$. The deep learning parameter predictor predicts the compartmental model parameters, and the compartmental model uses the predicted parameters to construct the predicted cumulative case curve for the pandemic.

HG-DCM operates as a two-stage pipeline: a deep learning component for parameter prediction, and a compartmental modeling component for incidence curve generation. The framework is defined by the following mapping:

$$\hat{\theta} = f(T, M) \tag{1}$$

$$\hat{y} = h(\hat{\theta}). \tag{2}$$

Here, the deep learning component, $f(\cdot)$, takes the observed time-series data ($T$) and epidemiological metadata ($M$) as input to predict the time-varying parameters ($\hat{\theta}$) of the underlying compartmental model. The subsequent component, $h(\cdot)$, maps these predicted parameters to the final forecast ($\hat{y}$) for the cumulative incidence curve by solving an Initial Value Problem (IVP). The key conceptual insight is that different pandemics share a common underlying mapping, $f$, between the observable early-stage dynamics ($T, M$) and the fundamental transmission parameters ($\theta$).

**Deep Learning Modeling** $f(\cdot)$    The function $f(\cdot)$ has two components: A Residual Network (ResNet) to extract embeddings from the historical pandemic time-series, and a fully connected component that concatenates the embeddings with epidemiological metadata to produce the final parameter prediction

The first ResNet component takes an input tensor of size $[L, N, D]$, where $L$ is the length of the time window, $N$ is the batch size, and $D$ (set to 1 for daily cases only) is the number of input features. Crucially, we made a structural modification to the standard ResNet: Batch Normalization (BN) layers were removed. Since the model is trained across historically distinct pandemics, the differences in batch statistics between past and current outbreaks introduce instability and prediction bias. Removing BN layers ensures the network relies solely on the learned weight distributions for generalization, improving robustness when generating embeddings.

Then, the generated embeddings are combined with epidemiological and demographic metadata (e.g., transmission pathways, healthcare expenditure). A full table of metadata is provided in Section A.1. The metadata are normalized using min-max scaling to the range $[0, 1]$, passed through two fully connected layers, and then concatenated with the time-series embeddings. The concatenated embedding is processed by final fully connected layers to produce the 12 parameters for the DELPHI model. To ensure the predicted parameters lie within their physical bounds (e.g., transmission rates are non-negative), we apply a sigmoid ranging function to normalize the output values.

**Compartmental Modeling** $h(\cdot)$    To generate the final forecast from the parameters, we utilize DELPHI (Li et al.), which is a highly ranked forecasting model on the COVID-19 Forecast Hub ensemble (Cramer et al., 2022) during COVID, as the compartmental model in this framework. DELPHI is a compartmental epidemiological model that extends the widely used SEIR model to account for under-detection, societal response, and epidemiological trends including changes in mortality rates. The model is governed by a system of ordinary differential equations (ODEs) across 11 states: susceptible ($S$), exposed ($E$), infectious ($I$), undetected cases who will recover ($U^R$) or die ($U^D$), hospitalized cases who will recover ($H^R$) or die ($H^D$), quarantined cases who will recover ($Q^R$) or die ($Q^D$), recovered ($R$) and dead ($D$). The transition rates between the 11 states are defined with 12 parameters, which we predict as $\hat{\theta}$ using the previous deep learning pipeline $f(\cdot)$. To generate the final incidence curve, the estimated parameters are passed through torchODE(Lienen & Günnemann), a parallel Initial Value Problem (IVP) Solver, to output the predicted cumulative case curve. We used Tsit5 with $a_{tol} = 1 \times 10^{-8}$, $r_{tol} = 1 \times 10^{-4}$ as the ODE solver. We refer the readers to (Li et al.) for details on the DELPHI model and its performance.

## 2.2    Data Processing and Augmentation

Given the data scarcity of historical outbreaks and the critical need for a model robust to noisy, early-stage data, we restrict training to daily cases and employ augmentation strategies.

**Data Preparation**    Before augmentation, we perform essential data cleaning:

- **Log Transformation**: Due to widely different case numbers across regions, we log-transform the case numbers for stability.
- **Exclusion**: Locations with fewer than 12 consecutive weeks of data where the cumulative case count was above 100 are excluded.

- **Missing Data and Noise**: Time series with weekly reporting or missing data are filled using linear interpolation for training only. Negative daily case counts are set to zero. No interpolation is used during model evaluation.

**Window-shift augmentation (past pandemics)**   To increase the sample diversity for past pandemics, we apply a window-shift augmentation technique. For each historical trajectory, we generate additional training samples by shifting the start of the input time series forward one day at a time. The augmentation stops when the input window's start date reaches the Last Day of Augmentation (LDoA), defined as the peak of the first epidemic wave. The LDoA is identified retrospectively in the historical data by: 1) computing and smoothing daily incidence (7-day centered rolling average), 2) detecting the first prominent peak (exceeding 25% of the global maximum using scipy.signal.find_peaks), and 3) defining the LDoA as the day of maximum incidence within that initial wave interval. Crucially, this retrospectively calculated LDoA is never used during inference on the current pandemic, preventing look-ahead bias and information leakage. Locations without a detectable first wave are excluded to ensure reliability. A graphical illustration of these three steps is shown in Figure 2.

**Masking augmentation (current pandemic)**   Because future observations are unavailable for an unfolding pandemic, we apply a masking strategy instead. Specifically, we apply a block-masking technique where a starting index is randomly selected within the input sequence, and the subsequent 7-day segment is replaced with zeros. This forces the model to learn robust temporal patterns even when contiguous data are missing.

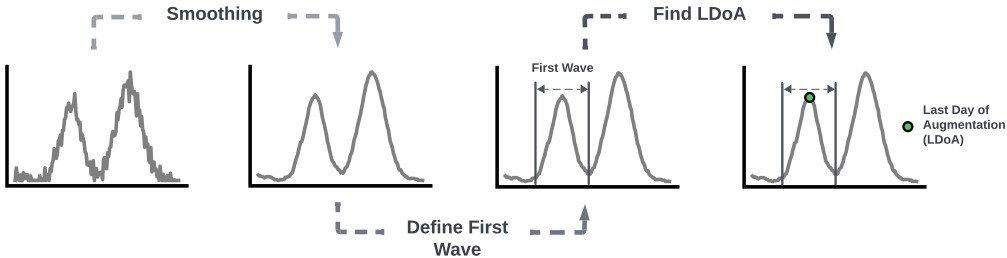

Figure 2: **Data Augmentation Methods** Window shift data augmentation method for past pandemic time series data

**Objective Function**   The objective function of HG-DCM is to minimize the loss between the predicted incidence curve and the actual incidence curve of past and current pandemics. The loss of past pandemics includes both the loss of the length-$t$ training window and the length-$v$ forecasting window (Eqn. 3). The current pandemic loss contains only the training window due to the lack of a forecasting window for an unknown future (Eqn. 4). Both losses of the past and current pandemics are calculated through a sum of mean absolute error (MAE) and mean absolute percentage error (MAPE) weighted by $\alpha$ to balance the effect of the population. The overall loss is calculated by a mean weighted by $\beta$ to balance between past pandemic losses and the current pandemic loss (Eqn. 5). The weight determines the amount of information inherited from past pandemics in predicting the current pandemic. Concretely, the formula for the loss function can be written as:

$$L_P = \frac{1}{n_P(t+v)} \sum_{i=0}^{n_P} \sum_{j=0}^{t+v} (|C_{ij} - \hat{C}_{ij}| + \alpha|\frac{C_{ij} - \hat{C}_{ij}}{C_{ij}}|) \tag{3}$$

$$L_C = \frac{1}{n_C t} \sum_{i=0}^{n_C} \sum_{j=0}^{t} (|C_{ij} - \hat{C}_{ij}| + \alpha|\frac{C_{ij} - \hat{C}_{ij}}{C_{ij}}|) \tag{4}$$

$$L = L_P + \beta L_C \tag{5}$$

where $n_p/n_c$ is the number of samples in the past/current pandemic data, and $C_{ij}/\hat{C}_{ij}$ is the actual/predicted cumulative cases of the $i$th pandemic at the $j$th time point.

## 3 EXPERIMENTS

### 3.1 EXPERIMENTAL SETUP

#### 3.1.1 DATA

We were unable to find a publicly available database that contained pandemic data from the past. Therefore, we constructed a pandemic dataset, which contains case and death (if available) time series data, pandemic meta-data, and country meta-data for major pandemic outbreaks and seasonal pandemics that have occurred worldwide since 1990. Only pandemics with significant (more than 100) and frequent (daily or weekly) reported incidences are included in the dataset. The dataset includes country-level and domain-level data on the following outbreaks: the 2020 COVID-19 pandemic, the 2014 Ebola pandemic, the 2003 SARS pandemic, the Peru (2000 - 2010) and Puerto Rico (1990 - 2008) Dengue Fever outbreak, and world-wide seasonal influenza outbreaks (2009-2023).

The time series dataset contains daily or weekly reported cases for each pandemic. The start date of pandemics differs for each location and is set by the first day when the cumulative case number exceeds 100. Epidemiological meta-data with uncertainties that were available at the early stage of the pandemic for each location are collected. The geological meta-data includes 13 country development indicators from the World Bank data (WorldBank) for each location in the dataset. The list of meta-data is available in A.1

#### 3.1.2 COMPARISON METHODS

We evaluate the model performances on early-stage forecasting tasks, where HG-DCM is used to forecast the cumulative case curve of 12 weeks based on 2/4/6/8 weeks of daily case data. To enable history-guided learning, HG-DCM is trained on a composite dataset of past pandemics, specifically Ebola, SARS, Dengue, and Seasonal Influenza, alongside the available early-stage data (2–8 weeks) from the current pandemic (COVID-19). Due to the lack of death data in pandemics prior to COVID-19, only case numbers are used to fit and evaluate the models in the experiments. Locations with no new daily cases reported during the training window are removed from the dataset.

For evaluation, we calculate the mean and median Mean Absolute Error (MAE) between the predicted and true cumulative incidence over the 12-week forecasting window. We compare HG-DCM against two advanced single-disease prediction models: GradABM (Chopra et al., 2023) and EiNNs (Rodríguez et al.). GradABM represents a differentiable agent-based modeling approach that leverages granular mobility and interaction data to simulate disease spread, offering high-fidelity simulations that differ fundamentally from compartmental approaches. EiNNs is a hybrid framework that fuses neural networks with epidemiological priors, designed to operationalize real-time signals for robust forecasting. Comparing HG-DCM against these distinct architectures—one agent-based and one hybrid—allows us to assess its effectiveness relative to the most capable current forecasting methodologies. The detailed setup of the baselines could be found in the appendix A.2. We attempted to benchmark HG-DCM against other models used for COVID-19 forecasting, specifically those included in the COVID-19 Forecast Hub (Cramer et al., 2022). However, most of these models lack publicly available, reproducible code bases, and the shared forecasting outputs do not include early-stage results (training windows $\leq$ 8 weeks), thereby limiting direct comparison. Moreover, the majority of models in the Forecast Hub are compartmental models.

To further understand the usefulness of each component of HG-DCM, we compare HG-DCM to its individual components, including DELPHI (Li et al.), and the Residual Convolution Neural Network (CNN) (Chung et al.). Through these ablation experiments, we aim to demonstrate that the HG-DCM architecture outperforms both stand-alone compartmental models and its component neural network.

### 3.2 RESULTS

#### 3.2.1 EARLY-STAGE FORECASTING BENCHMARKING

We first compare the forecasting accuracy of HG-DCM against GradABM and EiNNs across varying training window lengths (2, 4, 6, and 8 weeks) in Massachusetts and the United States. These locations were selected because they were the only locations in which there was available data and code for the comparison methods. As shown in Table 1, HG-DCM consistently achieves lower Mean Absolute

Table 1: Model Performance on Covid-19 Early Forecasting tasks in United States and Massachusetts. Locations are selected due to the limited data accessibility to run the comparison models. Dash indicates forecasting is not available in the specific location due to data constraints, see Appendix A.2. Bold indicates the best-performing models for each task.

| | United States | | | | Massachusetts | | | |
|---|---|---|---|---|---|---|---|---|
| | 2-Weeks | 4-Weeks | 6-Weeks | 8-Weeks | 2-Weeks | 4-Weeks | 6-Weeks | 8-Weeks |
| HG-DCM | **462,651** | 2,548,004 | **145,063** | **180,610** | **53,791** | **39,194** | 39,887 | **5,370** |
| GradABM | - | - | - | - | 245,682 | 231,213 | 188,082 | 142,934 |
| EiNNs | 801,152 | **729,091** | 496,680 | 295,222 | - | 46,097 | **25,669** | 10,874 |

Error (MAE) in most tasks compared to both baselines. Even though GradABM utilizes detailed mobility data to generate granular predictions, HG-DCM outperforms it by effectively leveraging priors from different pandemics to stabilize the trajectory, particularly in the 2-week and 4-week "cold-start" scenarios where data scarcity most severely impacts complex agent-based simulations. Similarly, while EiNNs superficially resembled our model by deep learning with epidemiological constraints, it is still limited to only considering the data from the current pandemic. HG-DCM's cross-disease transfer learning provides a more robust initialization, leading to significant error reductions in early-stage forecasts.

### 3.2.2 ABLATION STUDY

To isolate the contributions of specific components within our framework, specifically the roles of historical guidance, physics-based constraints, and deep learning architectures, we compare HG-DCM against three targeted variants: DELPHI, CNN, and T-DCM. The detailed model setups are available in Appendix A.2

Table 2: Model Performance on Covid-19 Early Forecasting. Bold indicates the best-performing models for each training window.

| | | 2 Weeks | 4 Weeks | 6 weeks | 8 Weeks |
|---|---|---|---|---|---|
| Mean MAE | | | | | |
| | CNN | 15600.4 | **11238.1** | 11012.5 | 10211.2 |
| | DELPHI | 342686.3 | 813807.8 | 29745.6 | 45140.7 |
| | T-DCM | **15049.2** | 17691.2 | 20571.1 | 24322.2 |
| | HG-DCM | 18602.6 | 110452.4 | **7112.5** | **4643.1** |
| Median MAE | | | | | |
| | CNN | 2963.4 | 2301.7 | **1187.8** | 871.8 |
| | DELPHI | 3609.1 | 2619.7 | 1249.2 | **537.7** |
| | T-DCM | 2745.8 | 2799.1 | 3101.0 | 4335.2 |
| | HG-DCM | **2231.1** | **1770.9** | 1275.6 | 796.0 |

**HG-DCM Outperforms DELPHI** Generally, both HG-DCM and DELPHI achieve higher accuracy as the length of training data increases. However, HG-DCM consistently outperforms DELPHI across forecasting horizons, particularly in the crucial early stages when only limited data are available. With 2 weeks of training data, HG-DCM reduces median MAE by 38.2% relative to DELPHI; with 4 weeks, the reduction is 32.4%. When 6 weeks of data are available, HG-DCM and DELPHI achieve comparable accuracy in terms of median error, but HG-DCM forecasts remain more stable across locations (Table 2, Figure 3). HG-DCM addresses a central limitation of compartmental models such as DELPHI, which is the tendency to overshoot case counts when trained on limited data. Overshoot arises from overfitting to the limited training data available, leading to forecasts that substantially deviate from observed trajectories. To formally quantify overshooting, we define it as occurring when the predicted cumulative case count in the final week of the forecasting window exceeds the corresponding observed value by more than fivefold. Across evaluation settings, HG-DCM exhibited markedly fewer overshooting events than DELPHI (Figure 4a). For example, in the case of the United States with an 8-week training window, DELPHI forecasts substantially overshoot true case numbers, whereas HG-DCM, by leveraging historical pandemic information, reduced overfitting and produced

predictions that were more consistent with real-world epidemic dynamics (Figure 4b). These results demonstrate the value of incorporating prior pandemic information to enhance early-stage forecasts when outbreak-specific data are scarce.

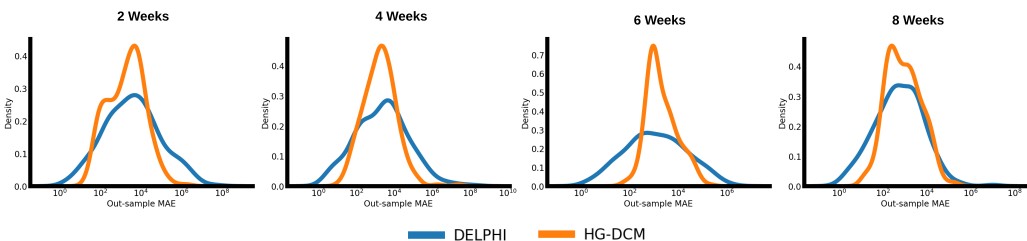

Figure 3: **Forecasting Window MAE Distribution** Forecasting window mean absolute error distribution for DELPHI and HG-DCM on COVID-19 12 Weeks Early Forecasting Tasks using 2 weeks, 4 weeks, 6 weeks, and 8 weeks of available data.

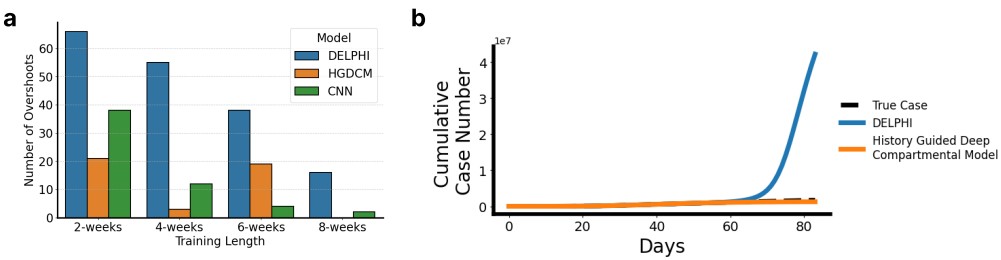

Figure 4: **Forecasting Example (a)** Number of overshooting predictions in different training window length for DELPHI, HGDCM, and CNN. **(b)** United States 8-week-training-window example where DELPHI suffers from overshooting caused by overfitting, while HG-DCM mitigates the overshooting by leveraging historical pandemic data.

**HG-DCM Outperforms End-to-end CNN Model** We next compare HG-DCM to a purely end-to-end CNN model. Unlike HG-DCM, which uses CNN to predict parameters of an epidemiologically grounded model, the CNN baseline bypasses mechanistic structure and directly predicts case trajectories from data. Despite its greater expressiveness, CNN generally underperforms HG-DCM across all training horizons. The performance gap is largest in the early stage (2–4 weeks of training data), where HG-DCM's integration of historical knowledge and compartmental dynamics yields markedly lower forecasting error (Table 2). These findings indicate that epidemiological inductive bias provides critical structure for learning, enabling HG-DCM to achieve both stronger predictive performance and greater interpretability than a black-box end-to-end model.

**HG-DCM Outperforms T-DCM** We further conducted an ablation study by training a Truncated Deep Compartmental Model (T-DCM) that excluded historical pandemic data and meta-data. The T-DCM was trained on datasets with 2, 4, 6, or 8 weeks of observations and evaluated on a 12-week forecasting task. Table 2 shows that T-DCM consistently underperformed HG-DCM across all training window lengths with respect to median MAE. Notably, HG-DCM achieved significant improvements in median MAE, with the gap widening as training data length increased. This result underscores the importance of incorporating historical context and structured meta-data for reliable forecasting in the early stages of pandemics.

Taken together, these results establish that HG-DCM effectively leverages historical pandemic data to guide compartmental modeling, producing more accurate and stable forecasts than both a leading compartmental model (DELPHI) and a purely data-driven end-to-end model (CNN). By combining mechanistic interpretability with neural network flexibility, HG-DCM represents a significant step forward in reliable early-stage pandemic forecasting.

### 3.2.3 PARAMETER INFERENCE

One of the key advantages of employing HG-DCM over traditional deep neural networks for pandemic forecasting is its interpretable parameterization. Unlike pure black-box models, the epidemiologically meaningful parameters predicted by HG-DCM can be extracted before being passed to the Initial Value Problem (IVP) solver for the compartmental model, which offers actionable insights.

To illustrate this advantage, we analyzed the parameters inferred by HG-DCM compared to the traditional DELPHI model in an early-stage COVID-19 forecasting task using four weeks of data (Figure 5). The DELPHI model's parameters exhibited a wide distribution, often leading to unstable forecasts and an overshooting problem. This instability arises because DELPHI fits models independently for each location, amplifying sensitivity to minor noise in the data. In contrast, HG-DCM leverages historical pandemic data and geospatial meta-data, ensuring more robust and consistent parameter estimation.

Statistical analysis using the *Wilcoxon Signed-rank Test* (Woolson) confirmed significant differences in all parameters, including the infection rate ($\alpha$), median day of action ($t_{\mathrm{med}}$), and rate of action ($r_s$), with $p$-values $< 0.05$. Specifically, HG-DCM predicted a lower infection rate, median day of action, and death rate, while exhibiting a higher rate of action. These findings suggest that, by adapting knowledge from past pandemics, HG-DCM avoids overfitting to the initial boost in case numbers and produces more conservative and realistic estimates, reducing biases that may otherwise arise from noise introduced by heterogeneous factors, including a lack of standardized case identification criteria in the early stage of data collection. The complete parameter analyses for all 12 DELPHI parameters can be found in Appendix A.4.

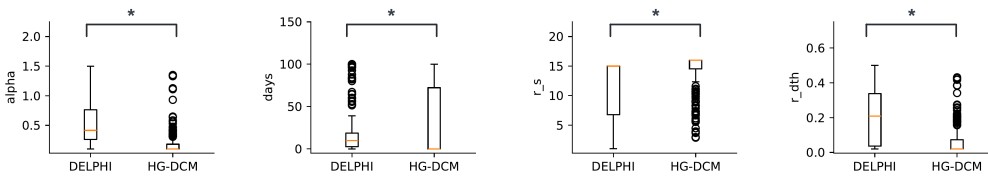

Figure 5: **Comparison of fitted parameters in DELPHI and HG-DCM models.** Box plots show the distribution of selected predicted parameters for DELPHI and HG-DCM. The central line represents the median, the box bounds the interquartile range, and whiskers extend to 1.5× IQR. Outliers are shown as points. Asterisks indicate statistically significant differences between methods (Wilcoxon signed-rank test, p < 0.05).

## 3.3 DISCUSSION

Our results demonstrate that HG-DCM consistently outperforms both standard compartmental models, such as DELPHI, and pure deep learning baselines during the early stages of a pandemic. The primary driver of this success is not the complexity of the neural network, but the strategic integration of historical data. In the "cold-start" phase, current data is often too sparse and noisy to effectively constrain the parameters of a differential equation. By introducing historical outbreaks as a source domain, we effectively increase the sample size from a few weeks at one location to months at hundreds of locations. This allows the model to learn robust priors—such as realistic ranges for infection rates and the typical shape of saturation curves, which stabilizes predictions when the current signal is weak.

Ultimately, these findings suggest a necessary shift in perspective for early-stage modeling: moving away from purely architectural complexity and toward data-centric generalization. While recent trends in deep learning favor increasingly large "black-box" models, our work indicates that in data-scarce environments, the diversity of the training signal is more critical than the depth of the network. By anchoring the flexible learning power of a neural network with the rigid, interpretable bounds of a compartmental model and the wisdom of historical data, we create a system that is robust against the overfitting that typically plagues standard approaches during the onset of a crisis.

## 4 LIMITATION

While HG-DCM demonstrates strong performance in early-stage pandemic forecasting, it is not without limitations. Our reliance on historical data also introduces granularity challenges. Unlike the high-resolution daily data available for COVID-19, older datasets such as Ebola and SARS were often reported in weekly aggregates. This necessitated the use of linear interpolation to align with our daily prediction framework. This approximation inevitably introduces errors, particularly during the volatile early stages of an outbreak where precise trend estimation is most critical.

Finally, HG-DCM is currently trained exclusively on confirmed case data, omitting mortality metrics. This decision was necessitated by the inconsistency and frequent unreliability of historical death records in past pandemics. While excluding this data allows us to maximize the volume of usable historical training samples, it limits the model's ability to jointly learn from case and death patterns.

## 5 CONCLUSION

In this study, we addressed the critical challenge of forecasting a new pandemic during its earliest stages, a time when data is scarce and standard models often fail. We introduced the History-Guided Deep Compartmental Model (HG-DCM), a framework that shifts the focus from building more complex architectures to making better use of available data. By treating historical pandemics such as seasonal Influenza, SARS, and Dengue as a source of knowledge, we demonstrated that deep learning models can learn universal patterns of disease spread and transfer them to a novel outbreak like COVID-19.

Our experiments on early COVID-19 data confirm that this history-guided approach significantly stabilizes predictions compared to methods that rely solely on the noisy, limited data of the current outbreak. We found that the inclusion of historical data acts as a powerful regularizer, preventing the model from overfitting to early fluctuations and guiding it toward more realistic epidemiological parameters.

Future work could focus on integrating additional data sources, such as mobility patterns, policy interventions, or other metadata, to further improve forecasting accuracy. Moreover, adapting HG-DCM for real-time applications represents an exciting avenue for research. We believe this work establishes a foundation for leveraging past pandemics through deep learning to inform future forecasting efforts.

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
