# A APPENDIX

## A.1 META-DATA

To incorporate epidemiological and geographical information into early-stage pandemic forecasting, we collected 7 epidemiological and 13 geographical metadata for each location and pandemic. Data unavailable during the early stages were marked as missing in the dataset. A detailed list of the metadata collected is provided in Table A.1.

Table A.1: **Meta data table for training HG-DCM** The meta-data table including the epidemiological metadata and geological metadata used in training HG-DCM

| Epidemiological Meta Data | Geological Meta Data |
| --- | --- |
| Basic Reproduction Rate (R0) | Population |
| Transmission Pathways | Net lending/borrowing |
| Mortality Rate | Current Health Expenditure per capita |
| Average Hospitalization Length | Population Density |
| Hospitalization Rate | GNI per capita |
| Latent Period | GDP per capita |
| Incubation Period | Physician per 1,000 people |
| | Urban Population Living in Slums |
| | GDP |
| | External Health Expenditure per capita |
| | Air Transport |
| | Government Effectiveness |
| | Domestic General Government Health Expenditure per Capita |

## A.2 BASELINES SETUP

**EiNN Setup** We utilized the official implementation of the EiNNs framework (Rodríguez et al.) as a hybrid baseline. To align the model with the specific early-stage forecasting windows defined in our experimental protocol, we adjusted the simulation's temporal configuration. Specifically, we shifted the earliest allowable start date from epiweek 202020 to 202014, allowing the model to capture the initial phases of the outbreak consistent with our dataset. Furthermore, to ensure a fair and direct comparison with HG-DCM, we modified the training objective. While the original implementation prioritizes mortality data, we altered the loss function to minimize error on confirmed case counts, matching the target variable and data availability of our proposed framework. 2-week forecasting for Massachusetts is not available for EiNN due to the model's minimum training sequence length requirement.

**GradABM Setup** For the GradABM baseline (Chopra et al., 2023), we employed the differentiable agent-based modeling framework to generate granular epidemiological projections. To strictly enforce consistency with the HG-DCM experimental setting, we adapted the optimization process to rely solely on case data. Consequently, we replaced the standard death-based loss function with a case-based loss function. This modification ensures that the model's calibration is driven by the same confirmed case signals available to HG-DCM, mitigating discrepancies arising from the lag or sparsity often associated with mortality data in the early stages of a pandemic. The available metadata of GradABM is only available for Massachusetts forecasting, and metadata generation scripts are not available. Thus, the forecasting results for the United States is not available for GradABM.

**HG-DCM Setup** Four HG-DCM models are trained using the 2/4/6/8-week training window respectively and predict for 12 weeks. Each HG-DCM is trained separately using the Adam optimizer with a stable learning rate of $1 \times 10^{-5}$. Given the large variation of case numbers among different locations, we use a customized geographic-pandemic sampler, where in each batch, one sample from each geographic-pandemic pair in the training data was sampled among all the augmentations. This approach accelerates the convergence by avoiding the turbulent loss curve caused by large variations in incidence numbers among different locations. Dropout or weight decay is not used when training

the model. The models are trained for 100 epochs, and the checkpoint with the lowest mean MAE for the training window of the current pandemic for each model is used in the comparison.

**Residual-CNN Setup**  We also train a ResNet-50 model for each training window using the same set of past pandemic data as the HG-DCM to prove that HG-DCM outperforms its component Neural Network. We utilize a learning rate of $1 \times 10^{-5}$ as optimized through a grid search. No dropout or weight decay is used.

**DELPHI Setup**  For fitting DELPHI models, the cumulative case curves are fitted separately for each location and each training window. Dual annealing (DA) (Xiang et al.) is used as the optimizer for parameter search. The same default parameter ranges as training HG-DCM are used to fit the DELPHI curve.

A.3  PARAMETER RANGES

To fit an accurate and interpretable model, it is crucial that the predicted parameters in both HG-DCM and DELPHI fall within epidemiologically reasonable ranges. For this study, we adopted the parameter ranges defined in the original DELPHI paper (Li et al.), with adjustments tailored specifically for early-stage forecasting. A comprehensive list of the parameter ranges used during model training is presented in Table A.2.

Table A.2: **DELPHI Parameters Ranges** The ranges of DELPHI parameters used in training HG-DCM and DELPHI

| DELPHI Parameters | Range |
|---|---|
| $\alpha$ | [0.1, 1.5] |
| days | [0, 100] |
| r_s | [1, 15] |
| r_dth | [0.02, 0.5] |
| p_dth | [0.01, 0.25] |
| r_dthdecay | [-0.2, 5.0] |
| k1 | [0.001, 5] |
| k2 | [0.001, 5] |
| jump | [0, 5] |
| t_jump | [0, 300] |
| std_normal | [0.1, 100] |
| k3 | [0.2, 2.0] |

A.4  COVID-19 EARLY-STAGE FORECASTING PARAMETER ANALYSIS

To better interpret the predictions, we analyzed the parameters inferred from HG-DCM compared to DELPHI in an early-stage COVID-19 forecasting task using four weeks of data. All 12 parameters are significantly different between DELPHI and HG-DCM based on Wilcoxon signed-rank test ($p < 0.05$). HG-DCM model tends to predict a lower $\alpha$, $days$, $r\_dth$, $t\_jump$, $p_d th$, $k1$, and $std\_normal$, whereas predicting a higher $r\_s$, $r_d thdecay$, $k2$, $junp$, $t_j ump$, and $k3$. The divergent prediction set of parameters from two different forecasting methods provides a more comprehensive understanding of the pandemic. (Figure A.1).

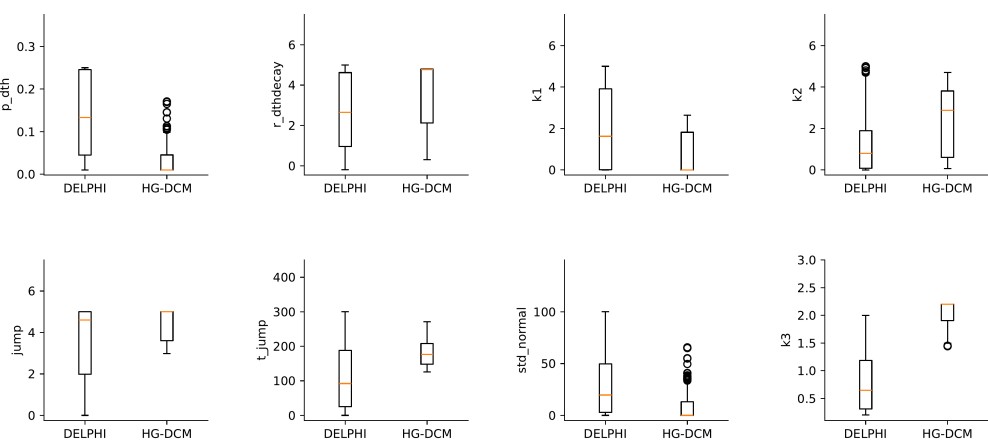

Figure A.1: **Comparison of fitted parameters in traditional DELPHI model and HG-DCM** The 8 bar graphs show the mean and standard deviation of the remaining 8 predicted parameters from two different approaches that are not shown in the main text. Mann-Whitney U test is used to calculate the p-value of the two groups. Pairs with p-values < 0.05 are considered significantly different.