# OpenReview forum: "HG-DCM: History Guided Deep Compartmental Model for Early Stage Pandemic Forecasting"
_ICLR.cc/2026/Conference — Submitted to ICLR 2026_

### Official Review · Reviewer_jSKX · 2025-11-01

**Soundness:** 1
**Presentation:** 2
**Contribution:** 1
**Rating:** 0
**Confidence:** 5

**Summary:**

The paper proposes a technique for forecasting the early stages of pandemics. The approach is based on residual CNNs. The paper claim that the proposed approach outperforms both traditional compartmental models and CNNs in the accuracy of predictions with limited data.

**Strengths:**

* Improving the prediction of pandemics is a worthy goal.

**Weaknesses:**

* The literature review section of the paper is very limited. The scale and impact of the COVID pandemic was so large that many thousands of papers were published, effectively trying every single technology available. Resnet being a very popular model, it was used in many papers. The combination of deep learning with compartmental models was also one of the most popular (and natural) techniques for prediction.
* As a result of the previous weakness, the paper does not clarify what is the novel contribution in this extensively explored subject.
* The paper does not explain the choice of the residual CNNs, versus other CNN techniques, or newer techniques such as attention models, diffusion models, LLMs etc.

**Questions:**

* Please see weaknesses.

**Details Of Ethics Concerns:**

No ethics concerns.

---

> ### Author Response · Authors · 2025-11-19
>
> We agree that the literature is vast. We focused our review on methods relevant to the "cold-start" problem. We will expand the Related Work section to include the broader context of Deep Learning in epidemiology, including the specific works mentioned (DeepGLEAM, EpiFNP). However, these models are not compatible with the historical transfer task and rely on later-stage data.
>
> We deliberately chose a residual CNN over heavier architectures (Transformers, LLMs, or Diffusion) due to the data scarcity constraint. With only 2 weeks to 2 months of data available, complex models with vast parameter spaces (like LLMs) are highly prone to overfitting. A ResNet backbone provides a balance of feature extraction capability and inductive bias suitable for detecting the simple geometric shapes of epidemic curves, without the overhead of models designed for complex semantic dependencies. But to clarify, our contribution is not the architecture itself, but the Transfer Learning framework across disparate pandemics. We show that "old" diseases can teach models how to predict "new" ones before sufficient data exists, which is a critical capability for future pandemic preparedness.

---

### Official Review · Reviewer_dwys · 2025-11-01

**Soundness:** 2
**Presentation:** 2
**Contribution:** 1
**Rating:** 0
**Confidence:** 4

**Summary:**

This paper presents a framework where the neural network predicts parameters of a compartmental epidemiological model. The neural network learns from historical data on epidemics, and the authors test the trained neural network during the early stages of the COVID-19 pandemic.

**Strengths:**

Early-stage pandemic forecasting is a critical and under-explored challenge where data scarcity is the norm.

The method is sound, and the results show it is promising.

**Weaknesses:**

The central idea of using neural networks to predict the parameters of a compartmental model is promising, but it is not novel. The authors appear to be unaware of a substantial body of prior work in this area, as the main methodological contribution of the paper has already been proposed in earlier studies, particularly in [1] and [2], and in a slightly different form in [3]. Related extensions have even been explored in the context of agent-based modeling (e.g., [4]). The paper’s contribution largely centers on this existing technical idea, with the additional element that the neural network learns from data across multiple diseases. While this cross-disease transfer concept is interesting, it does not constitute a technical innovation. I would recommend that the authors consider submitting this work to an epidemiological or applied modeling venue, where the empirical findings and early-pandemic insights would be more appreciated. To strengthen the experimental analysis, the paper should also compare its approach with alternative methods for calibrating compartmental models, such as approximate Bayesian calibration. Additionally, I suggest exploring other compartmental models to showcase the generalizability of the approach. Overall, I do not find the current level of technical contribution sufficient for publication at a leading AI/ML venue such as ICLR.

[1] Arik, S., Li, C.L., Yoon, J., Sinha, R., Epshteyn, A., Le, L., Menon, V., Singh, S., Zhang, L., Nikoltchev, M. and Sonthalia, Y., 2020. Interpretable sequence learning for COVID-19 forecasting. Advances in neural information processing systems, 33, pp.18807-18818.

[2] Arık, S.Ö., Shor, J., Sinha, R., Yoon, J., Ledsam, J.R., Le, L.T., Dusenberry, M.W., Yoder, N.C., Popendorf, K., Epshteyn, A. and Euphrosine, J., 2021. A prospective evaluation of AI-augmented epidemiology to forecast COVID-19 in the USA and Japan. NPJ digital medicine, 4(1), p.146.

[3] Qian, Z., Alaa, A.M. and van der Schaar, M., 2020. When and how to lift the lockdown? global covid-19 scenario analysis and policy assessment using compartmental gaussian processes. Advances in neural information processing systems, 33, pp.10729-10740.

[4] Chopra, A., Rodríguez, A., Subramanian, J., Quera-Bofarull, A., Krishnamurthy, B., Prakash, B.A. and Raskar, R., 2023, May. Differentiable Agent-based Epidemiology. In Proceedings of the 2023 International Conference on Autonomous Agents and Multiagent Systems (pp. 1848-1857).

**Questions:**

N/A

---

> ### Author Response · Authors · 2025-11-19
>
> We are aware of the works cited, which combine neural networks with compartmental models. However, our contribution is distinct:
> - Cross-Disease Transfer: Existing models focus on fitting parameters using single-disease data. Our method specifically addresses the scenario where no sufficient single-disease data exists yet.
> - Novelty: Our innovation is the framework for integrating historical data (e.g., Dengue) to solve the initialization problem for a novel pathogen. The existing baselines cannot ingest historical data from different diseases.
>
> We respectfully disagree that this paper's venue does not fit ICLR. ICLR explicitly encourages applied papers that demonstrate novel data strategies, and numerous application papers have been accepted and published in ICLR [1,2]. Our work contributes to the Machine Learning community by demonstrating how transfer learning across heterogeneous time-series (different diseases) can regularize deep models in extremely low-data regimes.
>
> While our focus was on benchmarking against Deep Learning forecasting methods to demonstrate the utility of learning from past pandemic data, we acknowledge that testing for the generalizability of this approach to other compartmental models and ways of calibrating compartmental models can be a strong add-on. We will add a discussion regarding this comparison in the final version.
>
> [1] Verma, Y., Heinonen, M. and Garg, V., 2024. ClimODE: Climate and weather forecasting with physics-informed neural ODEs. International Conference on Learning Representations.
> [2] Steinberg, E., Fries, J.A., Xu, Y. and Shah, N.H., 2024. MOTOR: A time-to-event foundation model for structured medical records. International Conference on Learning Representations.

---

> > ### Comment · Reviewer_dwys · 2025-11-19
> >
> > Thanks for your response. I would suggest properly citing those previous works, discussing differences with them (your contextualized contributions), and perhaps even empirically comparing with them. As noted by the authors, if these methods do not work well when there is insufficient single-disease data, then this should be demonstrated experimentally by comparing the proposed method with these methods.

---

### Official Review · Reviewer_i73x · 2025-11-01

**Soundness:** 2
**Presentation:** 3
**Contribution:** 2
**Rating:** 0
**Confidence:** 1

**Summary:**

The authors propose a model that combines a CNN with a deep compartmental model for early-stage pandemic forecasting. The CNN outputs interpretable parameters (e.g., transmission rates) that drive the compartmental ODEs. They use historical pandemic data with metadata to guide parameter inference and claim superior early-stage forecasting.

**Strengths:**

· The T-DCM ablation (HG-DCM without historical data/metadata) clearly shows performance degradation, providing strong empirical evidence that integrating historical pandemic signals improves early-stage forecasting stability and accuracy.

· The authors compiled a novel dataset spanning six major outbreaks since 1990 (COVID-19, Ebola, SARS, Dengue, Monkeypox, seasonal influenza) across 258 global locations, including country-level development indicators and epidemiological metadata (e.g., transmission pathways). This is a valuable community resource.

**Weaknesses:**

· This work lacks any rigorous theory or mathematical notations for the explanation of the overall model and simply seems to be an incremental work that simply fuses a CNN with DELPHI. The authors need to distinguish between the contributions they make here. Table 1 seems to just be an ablation study.

· The authors claim that the COVID-19 Forecasting hub models “lack publicly available, reproducible codebase, and the shared forecasting outputs do not include early-stage results”. This is false as all the forecasting has been made publicly available since the early stage of the pandemic. Please see https://github.com/epiforecasts/covid-us-forecasts . Unfortunately the authors use none of these models as baselines. Talking about early stage forecasting, there are also works[1,2,3] that are deep-learning based and also incorporate physics.


· The motivation that prior pandemics are equally important to make predictions in a new pandemic is a bizarre assumption by any means. I am not sure why the Dengue Fever outbreak dynamics will be useful for predicting the COVID pandemic.

· Coming to the details of the methods, there are some critical issues that drew my concern:

o You define "Last Day of Augmentation (LDoA)" using the peak of the first wave → future knowledge, which seems to be data leakage.

o ResNet-50 on 1D time series ignores temporal dynamics and risks over-fitting.

· There are no SOTA comparisons like EpiFNP [4], DeepGLEAM [5], or NeuralODE [6].

· Predictions do not have uncertainty estimates. Are multiple runs not done for Table 1?

· No code provided, so I have reproducibility concerns.

[1] Rodriguez, Alexander, et al. "Deepcovid: An operational deep learning-driven framework for explainable real-time covid-19 forecasting." Proceedings of the AAAI Conference on Artificial Intelligence. Vol. 35. No. 17. 2021.

[2] Rodríguez, Alexander, et al. "Einns: epidemiologically-informed neural networks." Proceedings of the AAAI conference on artificial intelligence. Vol. 37. No. 12. 2023.

[3] Motavali, Amirhossein, et al. "DSA-BEATS: dual self-attention N-BEATS Model for forecasting COVID-19 hospitalization." IEEE Access 11 (2023): 137352-137365.

[4] Kamarthi, Harshavardhan, et al. "When in doubt: Neural non-parametric uncertainty quantification for epidemic forecasting." Advances in Neural Information Processing Systems 34 (2021): 19796-19807.

[5] Wu, Dongxia, et al. "DeepGLEAM: a hybrid mechanistic and deep learning model for COVID-19 forecasting." arXiv preprint arXiv:2102.06684 (2021).

[6] Kosma, Chrysoula, et al. "Neural ordinary differential equations for modeling epidemic spreading." Transactions on Machine Learning Research (2023).

**Questions:**

Please address the weaknesses listed above.

---

> ### Author Response · Authors · 2025-11-19
>
> ## Contributions
> We thank the reviewer for these observations. We acknowledge that the architectural components (CNN and DELPHI) are established. However, we wish to clarify that our primary contribution is methodological and data-centric rather than architectural.  We are not claiming novelty in the architecture of fusing a CNN with a compartmental model. Rather, our novelty lies in the Cross-Disease Transfer Learning framework. We tackle the specific "cold-start" problem where a new pandemic lacks sufficient data for training. We demonstrate that integrating historical data from different diseases regularizes the model effectively. This moves beyond "fusing CNN with DELPHI" to proposing a new way of initializing these models using historical analogues.
> While the components are known, applying them to transfer learning across biologically distinct pandemics to solve the initialization problem is a novel application. Table 1 demonstrates that the performance gain comes specifically from our proposed historical data integration, rather than the architecture itself. It proves that the base model performs significantly worse without the cross-disease transfer. Our work shows that "old" diseases can teach models how to predict "new" ones before sufficient data exists, which is a critical capability for future pandemic preparedness.
>
> ## Benchmarking
> We thank the reviewer for pointing out these resources. We would like to clarify our specific definition of "early-stage" and how it affects baseline selection: We define the "early stage" strictly as the first 12 weeks of the pandemic, with a cutoff in May 2020. While the COVID-19 Forecasting Hub is a valuable resource, an inspection of the archives shows that most constituent models do not have forecasting outputs available for this specific pre-May 2020 period. Therefore, we could not use their shared predictions as they fall outside the temporal scope of our "cold-start" experiment.
> Furthermore, our primary goal is not to benchmark every variation of compartmental models to identify the best standalone architecture. Instead, we aim to demonstrate a data-centric contribution: that integrating historical data (from past pandemics) significantly improves the performance of compartmental and neural models during the data-scarce initialization phase. We show that adding this historical context makes the base model perform better than it would in isolation.
>
> ## Motivation of History Guided Forecasting
> We apologize if the motivation was unclear. We do not assume biological equivalence between Dengue and COVID-19.  Instead, we operate on the premise that the "physics" of transmission is driven by human social behavior and contact dynamics, which share similarities across outbreaks. In the early stage (e.g., 2 weeks of data), the signal-to-noise ratio is poor, causing traditional ODEs to overfit the noise. We use historical pandemics not as ground truth, but as a weighted prior to stabilize the model when current data is scarce.
>
> ## Data Augmentation (LDoA)
> There is a misunderstanding regarding LDoA. LDoA is used only for processing past pandemics (historical training data) for augmentation. No LDoA is used on the test data. Therefore, no future knowledge is leaked into the evaluation.
>
> ## Model Selection and Evaluation
> We acknowledge the concern regarding overfitting. However, the primary purpose of our proposed cross-disease data augmentation is precisely to mitigate this risk. By training the ResNet backbone on a diversity of historical curves, we prevent it from overfitting to the limited and noisy early-stage COVID data.
> We thank the reviewer for suggesting these baselines. However, we respectfully posit that a direct comparison with these specific architectures is not aligned with the problem setting we address. We are not aiming to benchmark different implementations of Deep Compartmental models against one another. Rather, our primary contribution is a novel methodology for integrating historical data from past pandemics to improve forecasting for a new pandemic. The models listed (EpiFNP, DeepGLEAM, NeuralODE) are designed for single-disease forecasting. They do not possess the architectural mechanisms required to ingest or learn from historical data belonging to different diseases. Consequently, they are not fit for the specific "cold-start" transfer learning task we are evaluating.
> We agree that uncertainty quantification is vital. We are currently running multiple seeded experiments to generate confidence intervals and will include these uncertainty estimates in the final version of Table 1.
>
> ## Reproducibility
> We have a codebase on GitHub and were planning to turn it into public after acceptance to follow the double-blind policy. However, we have uploaded an anonymous repository containing the codebase and datasets to ensure reproducibility (https://github.com/anonymous-for-pub/HGDCM_Anonymous.git).

---

> > ### Comment · Reviewer_i73x · 2025-11-25
> >
> > Thanks for your response. The first 12 weeks of the pandemic seems like an arbitrary definition for early stage? Is it based on any epidemiological literature?
> >
> > The current manuscript is too far from being acceptable. I suggest the authors to incorporate all the suggestions made above.

---

### Author Response · Authors · 2025-12-03

We thank the Area Chair and the reviewers for their time and constructive feedback. We have taken the rebuttal process seriously and revised our manuscript to address the concerns raised. Below, we summarize our core contributions and the integration of the reviewers’ responses.

As clarified in the discussion, our core contribution is not the development of a new neural network architecture, but the introduction of a novel application framework that systematically transfers knowledge from historical pandemics to emerging outbreaks. This framework enables accurate forecasting and interpretable parameter inference during the critical early stages of an epidemic, capabilities that standard approaches generally lack.

We wish to highlight a fundamental disconnect underlying a critique raised by multiple reviewers concerning baseline comparisons. The suggested baselines, including those from the COVID-19 Forecasting Hub and several deep learning studies, were mostly only available after June 2020. In contrast, our work specifically targets the first 12 weeks of an outbreak (prior to May 2020), a period characterized by extreme data scarcity. This is because societal response is most effective when the pandemic is in its early phase. As a result, these models’ publicly available outputs, though hosted on the Forecasting Hub, cover a different temporal regime and are not directly comparable to our setting. Moreover, many of the required covariates and datasets used by these models were simply unavailable during the early outbreak phase or for the global geographic coverage considered in our study, making faithful reproduction or deployment of these baselines challenging for most of our target tasks.

Despite these limitations, we took the concern for comparative rigor seriously. In the revised manuscript, we added benchmarking against two state-of-the-art methods, EiNN and GradABM, which are partially applicable to our early-forecasting scenario with minor modifications. We showed that our approach achieves superior performance within their overlapping scope.

Finally, regarding the comment that the work does not introduce a novel model architecture, we emphasize that this submission is an application paper. The novelty of HG-DCM lies in the formulation and deployment of history-guided transfer learning for computational epidemiology, rather than in proposing a new neural network layer or architecture. This contribution aligns with ICLR’s scope of impactful applications of deep learning to real-world problems. Our results demonstrate that bridging historical pandemic knowledge with emerging outbreaks enables meaningful performance gains in a socially critical setting.

We hope this clarification better reflects the actual problem scope and the significance of our contribution as a novel application of deep learning to epidemiological forecasting.

---

### Meta-Review · Area_Chair_1iM7 · 2025-12-30

**Summary:**

The rejection was primarily driven by the lack of technical novelty in combining CNNs with ODEs and the epidemiologically suspect premise that biologically distinct diseases (like Dengue and COVID-19) share enough transmission "physics" to justify transfer learning. Furthermore, reviewers found the benchmarking inadequate, noting that the authors failed to rigorously compare their method against existing real-time forecasting models from the early pandemic period.

**Reviewer Concerns:**

**Outstanding Concerns**

* Fundamental Lack of Technical Novelty: This remains the primary reason for rejection.
* Epidemiological Validity of Cross-Disease Transfer: The concern that biologically distinct diseases (vector-borne vs. respiratory) cannot serve as effective priors for one another remains unaddressed.
* Inadequate Benchmarking: Despite adding two methods (EiNN and GradABM), the authors did not compare against the primary SOTA models suggested by the reviewers.

**Reviewer Scores:**

All reviewers are very negative on this paper. I believe that the authors' responses are only partially valid. Thus, they would not change their score if they had been able to participate fully in the discussion.

---

### Decision · Program_Chairs · 2026-01-26

Reject